# A Novel Cone Model Filtering Method for Outlier Rejection of Multibeam Bathymetric Point Cloud: Principles and Applications

**DOI:** 10.3390/s23177483

**Published:** 2023-08-28

**Authors:** Xiaoyang Lv, Lei Wang, Dexiang Huang, Shengli Wang

**Affiliations:** 1College of Geodesy and Geomatics, Shandong University of Science and Technology, Qingdao 266590, China; 202183020044@sdust.edu.cn; 2Key Laboratory of Ocean Geomatics, Ministry of Natural Resources, Qingdao 266590, China; 3College of Ocean Science and Engineering, Shandong University of Science and Technology, Qingdao 266590, China; dexiang0314@sdust.edu.cn (D.H.); shlwang@sdust.edu.cn (S.W.); 4National Deep Sea Center, Qingdao 266237, China

**Keywords:** multibeam sonar systems, bathymetric point cloud, outlier rejection, Combined Uncertainty and Bathymetry Estimator, cone model filtering

## Abstract

The utilization of multibeam sonar systems has significantly facilitated the acquisition of underwater bathymetric data. However, efficiently processing vast amounts of multibeam point cloud data remains a challenge, particularly in terms of rejecting massive outliers. This paper proposes a novel solution by implementing a cone model filtering method for multibeam bathymetric point cloud data filtering. Initially, statistical analysis is employed to remove large-scale outliers from the raw point cloud data in order to enhance its resistance to variance for subsequent processing. Subsequently, virtual grids and voxel down-sampling are introduced to determine the angles and vertices of the model within each grid. Finally, the point cloud data was inverted, and the custom parameters were redefined to facilitate bi-directional data filtering. Experimental results demonstrate that compared to the commonly used filtering method the proposed method in this paper effectively removes outliers while minimizing excessive filtering, with minimal differences in standard deviations from human-computer interactive filtering. Furthermore, it yields a 3.57% improvement in accuracy compared to the Combined Uncertainty and Bathymetry Estimator method. These findings suggest that the newly proposed method is comparatively more effective and stable, exhibiting great potential for mitigating excessive filtering in areas with complex terrain.

## 1. Introduction

Multibeam sonar systems are widely utilized for underwater bathymetric surveys and target identification due to their full coverage and high efficiency. However, the presence of outliers, both including positive anomalies generated by instrument self-noise or environment noise and negative anomalies caused by seafloor secondary echo, significantly impairs the quality of multibeam bathymetric data. Therefore, it is imperative to eliminate these outliers. Usually, outlier rejection can be categorized into human-computer interactive (HCI) filtering and automatic filtering according to the level of human intervention. Although HCI filtering is highly accurate, it heavily relies on operator’s subjective awareness and experience, rendering it challenging to cope with massive amounts of multibeam bathymetric data. In light of this, exploring efficient and accurate automatic filtering methods has become a research focus in recent years. At present, two primary types of approaches are employed for effectively filtering multibeam bathymetric point cloud data (PCD), which rely on mathematical and statistical theories as well as function statistical inference values.

Earlier developed multibeam bathymetric data filtering methods based on mathematical statistics, such as the Ware method, Du method, and Median filter method [1,2,3], were deem to lack automation and accuracy [4]. Yang et al. proposed an automatic approach for outlier removal using beam point density and mathematical morphology methods, while also smoothing bathymetric data with wavelet filters [5]. This methodology effectively reduces echo detection errors and enhances the overall quality of the final bathymetric dataset.

Additionally, Mitchell presented a method of least-squares plane (also known as trend surface filtering, TSF) fitting multi-ping bathymetric data to filter noise [6]. The method is effective in rejecting obvious outliers, but it is less resistant to errors. Subsequently, Debese et al. proposed a hierarchical quadtree approach [7], while Rezvani et al. raised an M-estimation-based means for improving anti-aliasing and adaptivity respectively [8]. It is noteworthy that the Combined Uncertainty and Bathymetry Estimator (CUBE) method was first proposed by Calder and Mayer in 2003 [9], which employs Kalman filtering and multiple estimations to automatically process multibeam bathymetric data. Due to its high efficiency and error resistance, it remains one of the most advanced methods in this field. Even today, many scholars continue to study and improve this method, such as Zhao et al., who established a joint CUBE surface filtering parameter selection approach to effectively enhance the accuracy and efficiency of automatic processing of multibeam data [10].

Compared to filtering methods for multibeam bathymetric PCD, there have been more mature ways available for Airborne LiDAR PCD. For instance, various methods have been employed for Airborne LiDAR PCD processing, including using the difference in slope between adjacent points as a criterion for outlier rejection [11,12,13], employing mathematical morphology methods from images processing [14,15,16], utilizing irregular triangle networks [17,18], deep learning approaches for filtering [19,20,21], and applying physical model-based cloth simulation filtering (CSF) as demonstrated by Zhang et al. [22]. Given the many similarities between these two types of PCD, techniques suitable for filtering Airborne LiDAR PCD can also be applied to that produced by multibeam sonar. Based on this, Yang et al. used the quadratic surface Levenberg-Marquardt method to construct a transfer iterative trend surface and implemented a bi-directional fabric simulation filtering (BCSF) to eliminate the negative anomalies and solve excessive filtering [23]. Li et al. proposed an improved adaptive surface interpolation filter with multiple levels of hierarchy, utilizing a fabric simulation algorithm to generate effective initial ground seeds for constructing high-quality terrain surfaces [24]. Notably, Mahphood and Arefi applied a novel model inspired by tornadoes in nature to the ground point filtering of LiDAR point cloud [25], providing a new way for underwater multibeam point cloud filtering.

In general, the aforementioned methods can provide effective filtering in areas with relatively flat terrain. However, when confronted with complex and variable seabed terrain, it is challenging for one single method to achieve desirable filtering results [26,27,28]. Typically, varying degrees of excessive or incomplete filtering may occur. In this paper, a cone model filtering (CMF) method based on an improved tornado algorithm is proposed in attempt to address the above issues. The method mitigates the occurrence of excessive filtering in areas with intricate underwater terrain by introducing virtual grids and voxel down-sampling.

## 2. Description of the CMF Methodology

The core procedures of the CMF method primarily involve cone model construction, virtual grid segmentation, k-d tree data structure building and voxel grid division to select model vertices. Next, the principles of the method are described in detail.

### 2.1. Cone Model Construction

The cone used in the CMF method possesses an infinite number of tangent planes and satisfies the following equation:(1)$${\left(x-x_{v}\right)}^{2}+{\left(y-y_{v}\right)}^{2}={\left(z-z_{v}\right)}^{2}\times {\left(\tan\theta \right)}^{2}$$
where ($x_{v}$, $y_{v}$, $z_{v}$) denotes three-axis coordinate of the cone vertex, and *θ* denotes the half angle of the cone.

In a regular cone, each cross-sectional shape is a perfect circle, while each longitudinal section takes the form of an isosceles triangle. Therefore, the radius of any given transverse section can be accurately determined by considering both H (the height of the cone) and *θ*, as illustrated in Figure 1a.

To judge whether a point lies inside the cone, its horizontal distance from the axis of the cone must first be calculated using the left-hand side of Equation (1), and then compared with the radius of the cone at height z based on the right-hand side of Equation (1). If ${\left(x-x_{v}\right)}^{2}+{\left(y-y_{v}\right)}^{2}$ is less than or equal to ${\left(z-z_{v}\right)}^{2}\times {\left(\tan\theta \right)}^{2}$, as point p2 shown in Figure 1b, the point is deemed to lie inside; otherwise, it lies outside [25]. The CMF method dictates that points located inside the cone shall be deleted, while points situated outside of it shall be retained.

### 2.2. Virtual Grid Segmentation

In general, the actual seabed terrain is complex and variable, which may result in excessive or incomplete filtering if the CMF method is directly applied to the entire area. To preserve detailed information on raw data as much as possible and enhance CMF efficiency, this paper segments the raw point cloud into multiple regular virtual grids, as shown in Figure 2. The specific segmentation steps are outlined below:

Step 1. It is essential to ensure the availability of points in each virtual grid. The resolution of a virtual grid should ideally exceed that of the raw point cloud. A smaller GR value results in finer virtual grid segmentation.

Step 2. Determine the maximum and minimum values of raw PCD for both *x* and *y* dimensions, and calculate the number of columns and rows using Equation (2).
(2)$$\left\{ \begin{array}{l} N_{col}=\left[\frac{x_{\max}-x_{\min}}{GR}\right]+1\\ N_{row}=\left[\frac{y_{\max}-y_{\min}}{GR}\right]+1 \end{array}\right.$$
where “[ ]” denotes rounding down, $N_{col}$ denotes the number of columns, and $N_{row}$ denotes the number of rows. In particular, when the result of $x_{\max}-x_{\min}$ or $y_{\max}-y_{\min}$ in Equation (2) is exactly divisible by GR, “+1” is not needed.

Step 3. Compute the coordinates of each point in virtual grids according to Equation (3), and subsequently allocate all points to their respective grids.
(3)$$\left\{ \begin{array}{l} C_{i}=\left[\frac{x_{i}-x_{\min}}{GR}\right]+1\\ R_{i}=\left[\frac{y_{i}-y_{\min}}{GR}\right]+1 \end{array}\right.$$

*C_i_* and *R_i_* denote a column and row number, respectively.

Step 4. In an irregular area where the grid is void of any points, CMF filtering will not be executed.

### 2.3. K-D Tree Data Structure

The k-d tree is a commonly used data structure for efficient point retrieval in k-dimensional space [29,30]. In this study, a k-d tree was constructed to accelerate the spatial indexing of point cloud and improve computational efficiency. Figure 3 illustrates the principle of k-d tree construction.

The segmentation order is determined based on the variance of the point cloud in *x*, *y* and *z* dimensions. The hyperplane is then utilized to partition the space. For instance, as illustrated in Figure 3, the *x*-axis served as the initial segmentation dimension. The median is selected as the root node of the tree and utilized to divide the space into two parts with a red frame. Subsequently, the *z*-axis serves as the segmentation dimension, and further division of space is carried out using a green frame. Finally, by using the *y*-axis as another segmentation dimension, four subspaces are subdivided into eight subspaces with a blue frame to form leaf nodes. In this manner, recursive selection in each designated dimension yields both final leaf nodes and root nodes.

Once the k-d tree has been established, the subsequent search for neighboring target points follows the same building steps. In addition to locating the leaf node containing the target point from the root node, it is also necessary to traverse back to the root node in order to search for adjacent points. The search terminates when it is determined that no closer nodes exist.

### 2.4. Voxel Grid Division

The voxel grid approach is a widely adopted technique for down-sampling massive points. It utilizes the centroid to replace all points within each voxel grid, thereby preserving overall spatial structure information and ensuring uniform distribution of sampling points [31,32].

Firstly, the virtual grid that has already been segmented is further subdivided into smaller voxels, as illustrated in Figure 4a,b. Voxel size should be chosen to be larger than the resolution of the raw PCD and smaller than the GR, ensuring that each voxel grid contains at least one point. Equation (4) is used to calculate the centroid of all points within each voxel grid. The red point in Figure 4c is the calculated centroid of points in each voxel grid.
(4)$$\left\{ \begin{array}{l} x_{centroid}=\frac{\sum_{i=1}^{n}x_{i}}{n}\\ y_{centroid}=\frac{\sum_{i=1}^{n}y_{i}}{n}\\ z_{centroid}=\frac{\sum_{i=1}^{n}z_{i}}{n} \end{array}\right.$$
where ($x_{centroid}$, $y_{centroid}$, $z_{centroid}$) denotes centroid coordinate of voxel, *n* denotes number of points in a voxel.

The size of the voxel grid should be determined based on the selected *θ* angle. For example, when the terrain is flat and the *θ* angle is large, as shown in Figure 5a, increasing the voxel size may be appropriate. Conversely, as depicted in Figure 5b for sloping terrain with a decreasing *θ* angle, reducing the voxel size accordingly is necessary to ensure less incomplete filtering.

## 3. Experimental Section

This section presents an overview of bathymetric data acquisition and pre-processing procedures, followed by a detailed description of PCD processing with the CMF method.

### 3.1. Bathymetric Data Acquisition and Pre-Processing

The data utilized in this study was obtained from a project conducted in the sea region of Zhejiang Province, China, as depicted in Figure 6a. The multibeam bathymetric data were acquired using a Reason SeaBat T50-P shallow-water multibeam sonar system with a beam opening angle set at 120 degrees and containing 512 beam points per ping. Navigation and positioning system were facilitated by the POS MV OceanMaster from Applanix Canada, while real-time sound velocity profiles were collected using the MTNOS X SVPT from AML Canada. GNSS tide measurement was used to obtain accurate dynamic tidal levels [33]. The water depth of the whole measurement area ranged from 2.40 m to 53.90 m, as shown in Figure 6b.

The acquired multibeam bathymetric data was pre-processed using CARIS HIPS 11.3 software, which includes sound velocity correction, installation deviation calibration, navigation attitude correction and tidal level correction. Subsequently, the raw PCD was outputted and four representative areas were selected for detailed depiction, including Area 1 characterized a flat terrain as shown in Figure 7a, Area 2 featuring a sloping terrain as shown in Figure 7b, Area 3 comprising a mixed terrain as shown in Figure 7c and Area 4 containing a steep terrain in Figure 7d. The mixed area encompasses flat, sloping and locally steep terrain, representing a common complex seafloor topography. Collectively, Area 1 comprises 75,283 bathymetric points; Area 2 comprises 59,799 bathymetric points; Area 3 comprises 186,660 bathymetric points; and Area 4 comprises 85,005 bathymetric points.

### 3.2. Point Cloud Filtering

To validate the efficacy of the CMF method, four filtering approaches, namely CUBE, TSF, CSF, and CMF were employed to filter the raw PCD. The implementation of these four methods were carried out using Qt in conjunction with Point Cloud Library (PCL). The experiments were conducted on version 5.12.9 of Qt and version 1.8.1 of PCL. This section primarily elucidates the workflow of the CMF method as shown in Figure 8.

Step 1. Statistical filtering method was used for spatial analysis of point cloud to remove large-scale non-ground points. The mean and standard deviation can be calculated using Equation (5) and Equation (6), respectively.
(5)$$\mu =\frac{\sum_{i=1}^{N}D_{i}}{N}$$
(6)$$\sigma^{2}=\frac{\sum_{i=1}^{N}{\left(D_{i}-\mu \right)}^{2}}{N}$$

By setting the standard deviation multiplier *n*, any points falling outside the range “*μ* ± *n* × *σ*” were identified as noise and eliminated. In order not to filter out sparse real ground and edge terrain, the *n*-value of the statistical filtering here should be set large.

Step 2. Proceeded to determine the maximum extent of the PCD after statistical filtering. Subsequently, divided the virtual grid based on PCD resolution and allocate corresponding PCD to its respective grid.

Step 3. The point cloud within each virtual grid was organized into a k-d tree structure and again statistical filtering was applied. The statistically filtered results were used to calculate the slope angle. The *n*-value should be set small here in order to improve the accuracy of the angle calculation. The slope angle of each point relative to its eight nearest neighbors was calculated after statistical filtering. In the virtual grid, the maximum of these slope angles was chosen as the *β* angle.

In this study, the angle *β* was classified as follows: when it is greater than 0 degrees but less than 20 degrees, it is identified as a flat terrain; when is greater than or equal to 20 degrees but less than 50 degrees, it is recognized as a sloping terrain; otherwise, it is considered as a steep terrain. All three types of terrain (flat, sloping and steep) were selected to determine the threshold of *θ* angle—a crucial parameter in the cone model—whose range can be determined using Equation (7).
(7)$$\theta <90^{\circ }-\beta$$

Step 4. Down-sampled the PCD in voxel grids to determine the vertices of the model. Certainly, the number of model vertices should be determined considering the size of the selected *θ* angle.

Step 5. Judged whether a point lies inside the model. As dictated, a point would be removed as a non-ground point if D1 was less than or equal to D2.

Step 6. Applied cone models in one virtual grid to reject non-ground points, followed by traversing other virtual grids until all PCD judgments were completed.

It should be noted that through the procedures above only the positive anomalies can be eliminated. To remove negative anomalies deeper than the actual seafloor topography, the PCD needs to be flipped and steps 3—step 6 be repeated to realize bi-directional CMF filtering.

## 4. Results and Discussion

This section encompasses two aspects. Firstly, a viable range of model parameters was derived through two sets of comparative experiments, involving the selection of different *θ* angles for a fixed voxel size and the selection of different voxel sizes for a fixed *θ* angle. Secondly, the filtering results obtained from four methods (CUBE, TSF, CSF and CMF) were comparatively analyzed across three different terrain types (flat, sloping and mixed).

### 4.1. Selection of Model Angles

The evaluation standard for filtering accuracy employed a method that integrated Type I error, Type II error, and Total error. During the experiment, a voxel size of 1 × 1 × 1 m was selected, and filtration efficiency was measured at different angles. The experimental findings are presented in Figure 9.

In Figure 9, it can be observed that the curves representing Type I error closely aligned with the total errors when comparing error results for the three area types. This is due to a significantly smaller number of non-ground points compared to ground points. As the selected *θ* angle increased, more internal points of the cone were eliminated, resulting in a decrease in Type II error.

As depicted in Figure 9a, within a flat terrain, the Type II error gradually diminished with an increasing *θ* angle due to the model’s enhanced capability in eliminating non-ground points. When the *θ* angle was below 60 degrees, there was minimal occurrence of Type I error. However, when it exceeded 60 degrees, there was a noticeable increase in Type I error as genuine ground points were mistakenly filtered out by the model. Henceforth, it is advised that the *θ* angle should not exceed 60 degrees to avoid excessive filtering for flat terrains.

As shown in Figure 9b, maintaining Type I error was more effective when the *θ* angle was kept below 40 degrees for sloping area. Once it exceeded this threshold, the error rate started to increase due to excessive filtering of sloping ground points. On the other hand, Type II error decreased as *θ* angle increases and dropped below 30% when *θ* angle surpasses 40 degrees. Therefore, it is suggested that the *θ* angle should not exceed 40 degrees to avoid excessive filtering for sloping terrains.

As illustrated in Figure 9c, the steep terrain was more sensitive to the change in *θ* angle than the other two terrains. As the *θ* angle increased, the Type I and Type II errors showed more variation, especially the Type I error. This is because the *β* angle of the steep terrain was very large, and as the angle of the model continues to increase, it will lead to the deletion of more sloping ground points, i.e., the phenomenon of excessive filtering. Therefore, it is recommended that the *θ* angle should not exceed 20 degrees to avoid excessive filtering for steep terrain.

### 4.2. Determination of Voxel Size

The vertex is another crucial parameter of the cone model. In this section, we employed the voxel grid method to select the vertices of the cone model and conducted a comparative analysis in a mixed area. To verify the relationship between *θ* angle and voxel size, we fixed *θ* angles at 20 degrees, 40 degrees and 60 degrees, respectively. Subsequently, different voxel sizes were selected for quantitative analysis of the filtering results. The experimental results are shown in Table 1.

As shown in the table, the trends of Type I error and Type II error varied with voxel size. Specifically, Type I error was inversely proportional to the voxel size; decreasing voxel size led to an increase in Type I error, particularly for larger *θ* angles. This can be attributed to the fact that selecting more vertices at larger model angles may result in removal of real ground points around those vertices, thereby increasing the likelihood of a Type I error. It was worth noting that the smallest Type I error occurred when the *θ* angle was 20 degrees and the voxel size was 1 × 1 × 1 m. This can be described to the similarity between the resolution of the raw PCD and the selected voxel size, resulting in a sparse point distribution within each voxel grid. Additionally, due to an insufficiently small *θ* angle for the experimental area, the outliers were not effectively filtered.

On the contrary, if the *θ* angle was small and the selected model vertices were equally few, the spacing between the two models was large and the outliers in the middle part would not be removed. According to the results in the table, with the increase of voxel size, the Type II error increased, and the filtering effect decreased significantly. Therefore, it is suggested that more vertices should be selected in the case of smaller *θ* angle. This will eliminate noise and will also suppress the excessive filtering phenomenon.

In addition, the results presented in Table 1 indicated that the execution time of CMF experienced a slight decrease when the voxel size increased from 1 to 5. This outcome can be attributed to the fact that with an increase in voxel size, there was a reduction in the number of model vertices and frequency of judgments made, leading to a corresponding decrease in time. However, this change was not significant owing to fewer point cloud being present within the selected area.

In summary, the number of selected vertices is inversely proportional to the size of the voxel grid, while the *θ* angle is inversely proportional to the number of selected vertices. For flat terrains, it is recommended to set a large *θ* angle for effective filtering and reduce the voxel size in order to avoid filtering out more ground points near the vertices, thus improving operational efficiency. For sloping and steep terrains, increasing the selection number of vertices and decreasing the *θ* angle are necessary to ensure accurate filtration.

### 4.3. Performance Comparison of Different Filtering Methods

#### 4.3.1. Qualitative Analysis

The filtering results obtained through four distinct methods (CUBE, TSF, CSF and CMF) in three different areas, namely Area 1, Area 2 and Area 3, are illustrated in Figure 10, Figure 11 and Figure 12, respectively.

In Area 1, all four filtering approaches demonstrated satisfactory results. However, the CUBE method exhibited limited effectiveness in removing a few large-scale points at the edges of Figure 10a. The TSF method displayed localized areas of excessive filtering and was unable to adequately filter out small-scale outliers at the edges of Figure 10b. As depicted in Figure 10c, the CSF method effectively eliminated positive anomalies, it still left behind a small fraction of negative anomalies and may result in excessive filtering. In contrast, the CMF method exhibited no signs of excessive filtering and achieved superior noise reduction as shown in Figure 10d. The comparative analysis revealed that by increasing the *θ* angle and reducing vertex numbers the CMF method can ensures a decrease in excessive filtering while maintaining optimal performance.

In Area 2, the CUBE method exhibited incomplete filtering as shown in Figure 11a. While both the TSF method as shown in Figure 11b and the CSF method as shown in Figure 11c demonstrated excessive filtering, the latter achieved a more optimal filtering effect. The former showed the most obvious degree of excessive filtering, as the fitted trend surface did not reflect the complex seafloor topography well. The CMF method achieved the optimal filtering results as shown in Figure 11d due to its refined parameter selection strategy that effectively reduced *θ* angle and increased vertex numbers.

In Area 3, the CUBE method achieved the smoothest surface of the seafloor topography as depicted in Figure 12a. However, excessive filtering occurred where there was high variability in the topography, leading to a loss of some smaller geomorphic features. According to the filtering results of the TSF method as depicted in Figure 12b and the CSF method as illustrated in Figure 12c, it can be seen that both methods showed different degrees of excessive filtering due to the limitations of the surface fitting function and fabric stiffness. Moreover, the former also exhibited incomplete filtering phenomenon. In contrast, the CMF method can not only effectively remove most of outliers but also preserve genuine topographic features by reducing *θ* angle and voxel size as shown in Figure 12d.

#### 4.3.2. Quantitative Analysis

The methods presented in this paper were quantitatively analyzed by comparing the results obtained from HCI filtering, which were taken as true values, with the four methods for assessing mutual differences. The summarized findings can be found in Table 2.

In Area 1, the results reveal that all four methods exhibited favorable overall performance. Among them, the TSF and CSF method showed a max-depth deviation of approximately 2.1 m from the true values due to a few remaining outliers. Combined with Figure 10b,c, it can be seen that there were small-scale anomalies that were not filtered out in both methods, and a small degree of excessive filtering occurred in the CSF method. Therefore, the standard deviation of CSF differed the most among the four methods with 0.0053 compared to the true values. The standard deviation obtained by the CUBE method was similar to that of the TSF method, but as seen in Figure 10c, there were still large-scale outliers that were not removed. The CMF method did not show excessive filtering and had the smallest standard deviation of 0.0012 from the true value.

In Area 2 and Area 3, there was a large gap between the experimental results obtained by the TSF method and the true values, and the difference of standard deviation was 0.2495 and 0.2330, respectively. Because the TSF method was plagued by uncertainties in surface fitting function, incomplete filtering and the unreasonable removal of some water depth points [34]. As a result, in Figure 11b and Figure 12b, the method appeared a large degree of excessive filtering and incomplete filtering. In addition, the standard deviation obtained by the CSF method was closer to the true values. However, in Area 3, the max-depth obtained by the method differed from the true value by 4.44 m, with an average deviation of about 0.099 m. Combined with Figure 12c, it can be seen that the method was less effective in filtering negative anomalies and there was obvious excessive filtering.

In Area 2, the results obtained from the CUBE and CMF methods were comparable, and the differences between the two methods and the true values were small. However, for complex terrain, the CUBE method may remove more of the true seafloor topography [35]. As shown in Figure 12a, there was excessive filtering in the CUBE method with a standard deviation of 0.0187 from the true value. In contrast, the CMF method showed the smallest variance deviation from the true value in Area 2 and Area 3, which proved the stability and reliability of the results. However, in Area 3, the maximum depth obtained by the CMF method differed from the true value by 0.44 m, because some small-scale anomalies were not removed, as shown in Figure 12d.

To ensure accurate comparison between the two methods, we utilized the latest standard published by the International Hydrographic Organization (IHO) [36]. The results are presented in Table 3.

As indicated in Table 3, the CMF method demonstrated improved accuracy compared to the CUBE method across various areas. Notably, the disparity between these two methods was minimal in Area 1 and Area 2, with differences of merely 0.36% and 0.16%, respectively. However, in Area 3, the CMF method exhibited a significant enhancement of 3.57% over the CUBE method, thus validating the efficacy of our proposed approach.

## 5. Conclusions

This paper proposes a novel CMF method based on an enhanced tornado algorithm for filtering multibeam bathymetric data. The core procedures of the proposed method mainly include four aspects, which are cone model construction, virtual grid segmentation, k-d tree data structure building and voxel grid division, respectively.

The cone model is primarily characterized by its angles and vertices, with the constraint that the sum of the model angle and maximum terrain slope should not exceed 90 degrees to prevent excessive filtering. Based on experimental findings, it is recommended to restrict the *θ* angle to 60 degrees for flat terrains to avoid excessive filtering. For sloping terrains, a suggested upper limit of 40 degrees for the *θ* angle is advised. In case of steep terrains, it is highly advisable not to surpass a *θ* angle of 20 degrees. The voxel size is correlated to the resolution of the PCD, and it has been observed that the model angle tends to be inversely proportional to the number of vertices.

The results demonstrate that the CMF method exhibits the smallest standard deviation when compared to the actual value through comprehensive comparisons across different areas and methods. These findings not only highlight the effectiveness and stability of the CMF method but also underscore its significant potential in addressing excessive filtering in complex terrain. In terms of future development, it is suggested to consider incorporating self-adaptive adjustment of model parameters and enhancing automation and precision by improving the accuracy of *β* angle calculations.

## Figures and Tables

**Figure 1 sensors-23-07483-f001:**
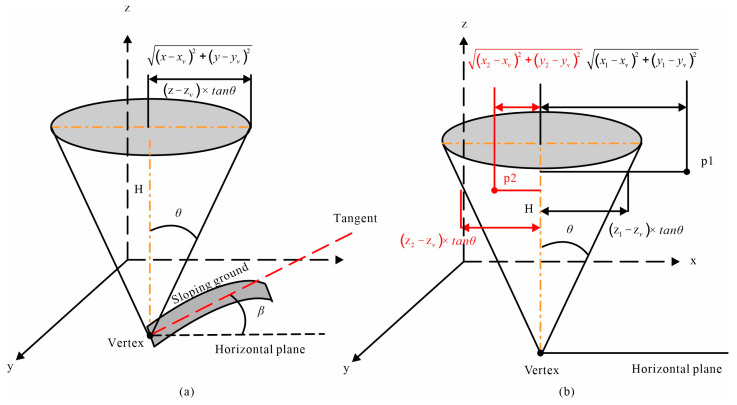
Cone model and its filtering principle. (**a**) Inverted cone model and its primary elements; The red dotted line is tangent to the sloping ground, and *β* denotes the maximum slope angle of local terrain. (**b**) Schematic diagram of outlier rejection principle.

**Figure 2 sensors-23-07483-f002:**
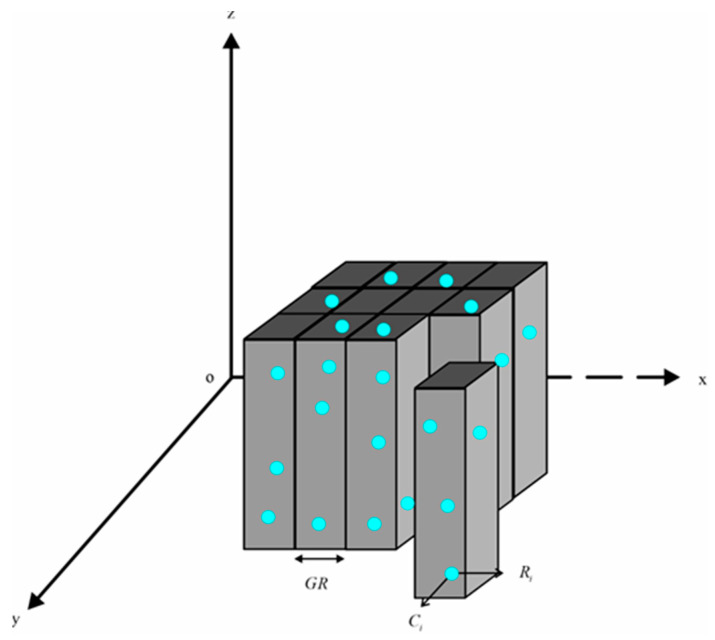
Principle of virtual grid segmentation. GR denotes grid resolution. The blue dots denote the raw PCD.

**Figure 3 sensors-23-07483-f003:**
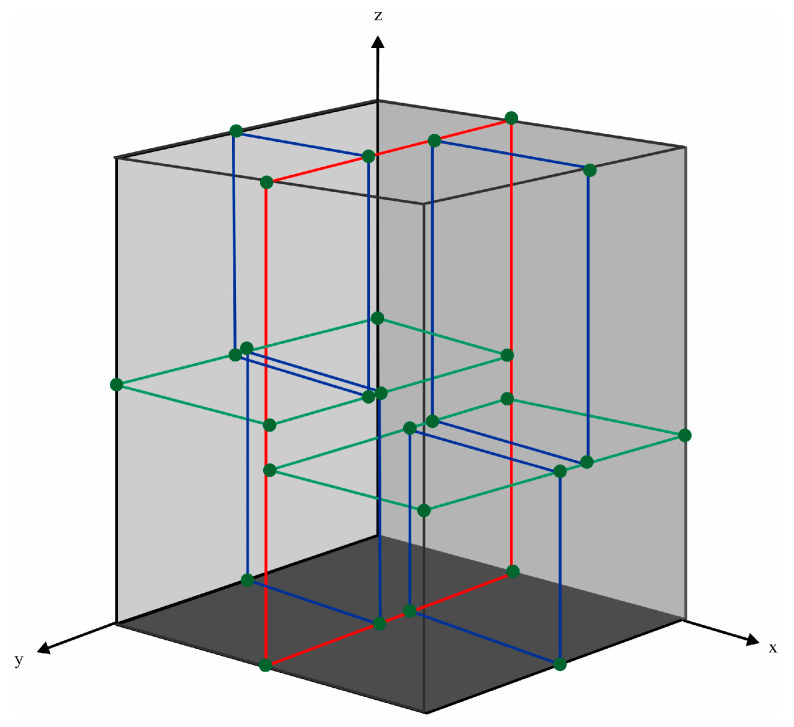
The k-d tree data structure based on the principle of hyperplanes, represented by red, green and blue frames.

**Figure 4 sensors-23-07483-f004:**
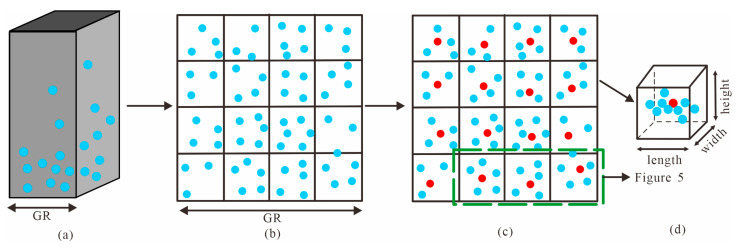
Voxel down-sampling in a virtual grid. The blue dots denote the raw PCD. (**a**) Raw points in each virtual grid; (**b**) Subdivision of virtual grid into voxel grids (Top view); (**c**) Determination of centroid in each voxel grid (Red points); (**d**) The shape of each voxel grid is a regular cube.

**Figure 5 sensors-23-07483-f005:**
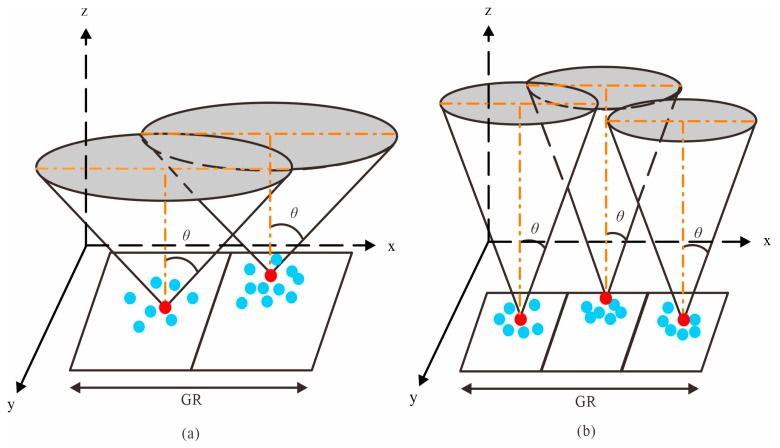
Principles of voxel grid size selection. The blue dots represent the raw PCD, while the red dots represent the centroid of all point clouds in a voxel grid. The horizontal color lines indicate the diameter of the base of the cone, and the vertical color lines represent its height. (**a**) Voxel grid size selection strategy for flat terrain; (**b**) Voxel grid size selection strategy for sloping areas.

**Figure 6 sensors-23-07483-f006:**
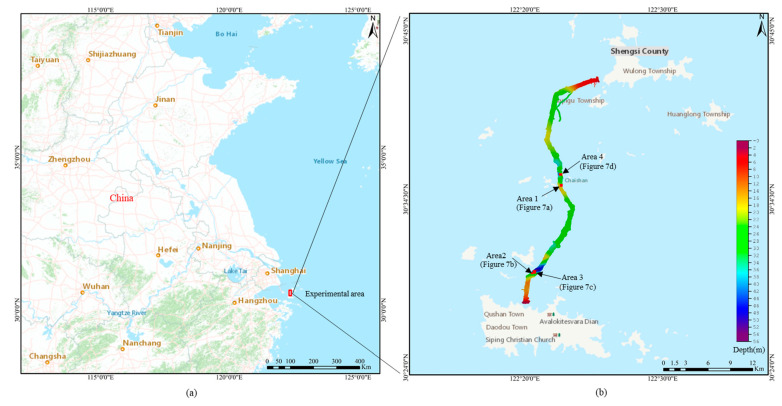
Experimental area of this study. (**a**) Location of the experimental area; (**b**) Bathymetric map of the study area.

**Figure 7 sensors-23-07483-f007:**
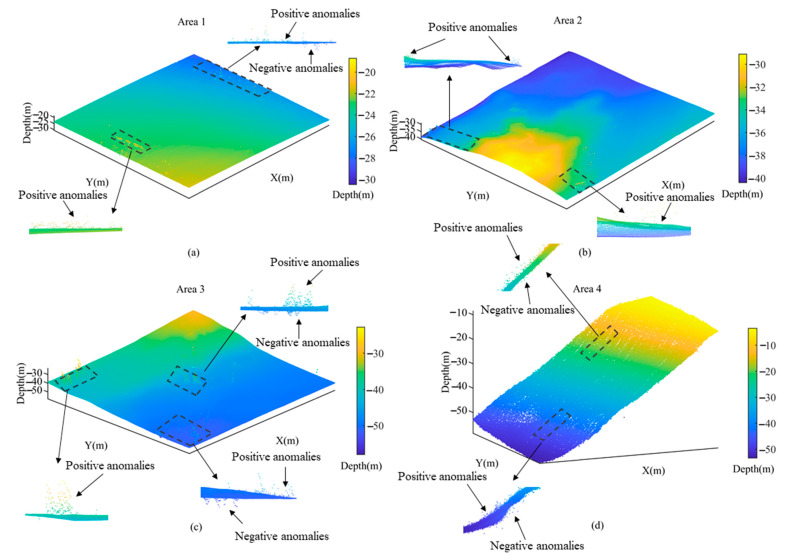
Four representative areas selected from the experimental sea region. (**a**) Area 1 characterized by a flat terrain; (**b**) Area 2 featuring a sloping terrain; (**c**) Area 3 comprising a mixed terrain; (**d**) Area 4 containing a steep terrain.

**Figure 8 sensors-23-07483-f008:**
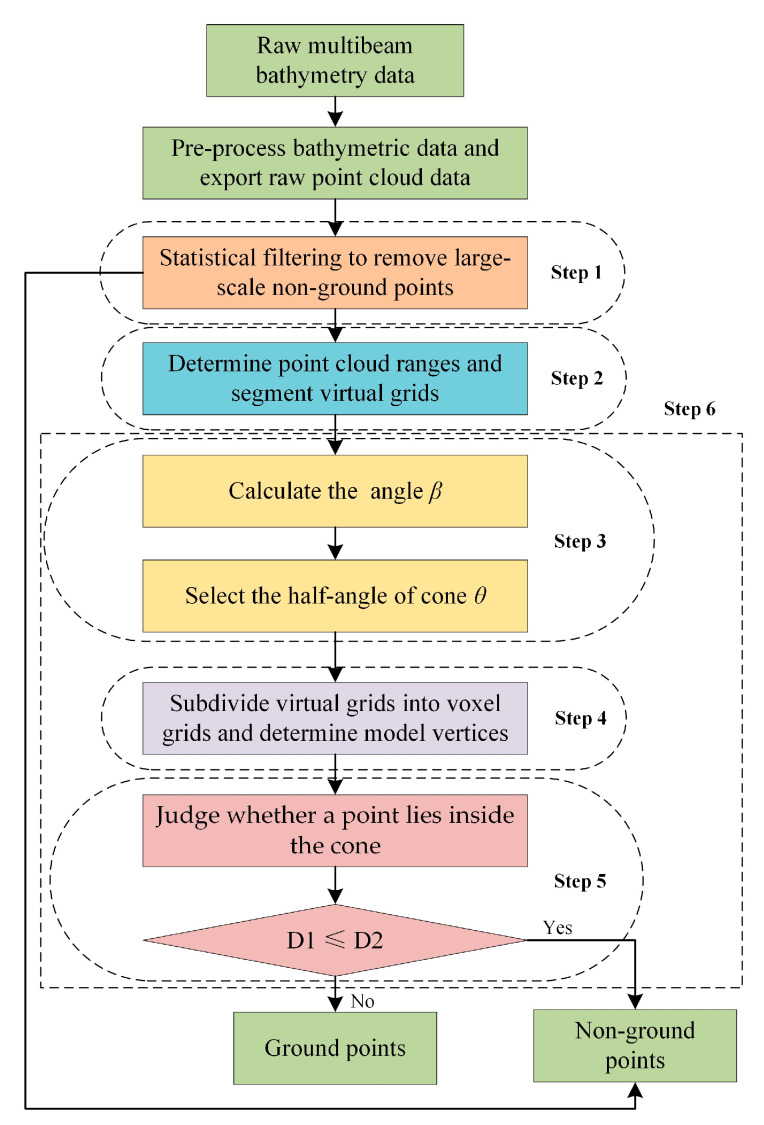
CMF flow chart. D1 represents the horizontal distance between a point and the central axis of the cone, while D2 denotes the radius of its transverse section at that corresponding height in the model.

**Figure 9 sensors-23-07483-f009:**
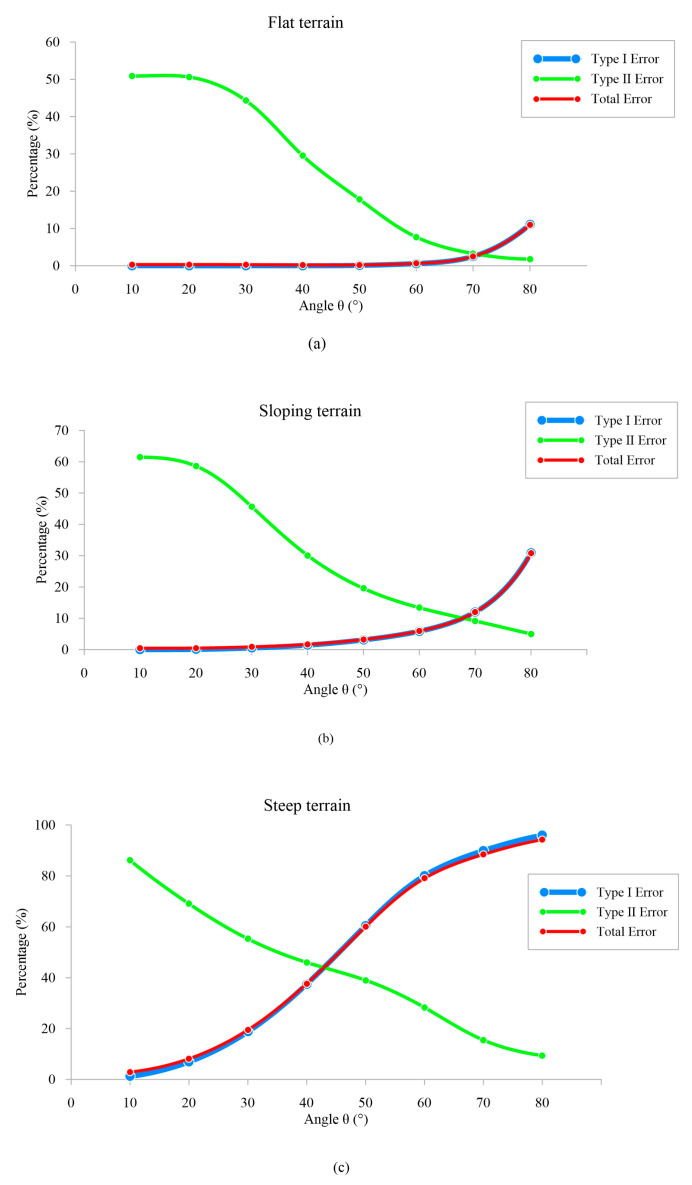
Trends of different types of errors in various terrain and angles, compared with HCI results. Type I error represents misclassification of ground point as a non-ground point; Type II error represents misclassification of non-ground point as a ground point. (**a**) Error curves for flat terrain; (**b**) Error curves for sloping terrain; (**c**) Error curves for steep terrain.

**Figure 10 sensors-23-07483-f010:**
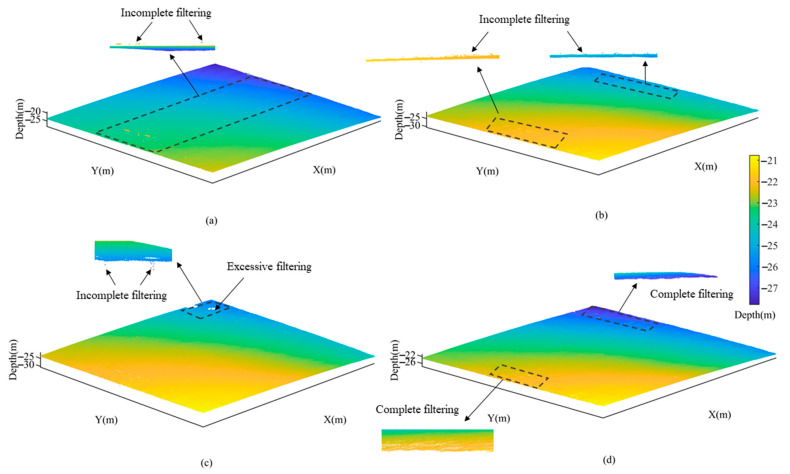
Comparison of filtering results in Area 1 using different methods. (**a**) CUBE; (**b**) TSF; (**c**) CSF; (**d**) CMF.

**Figure 11 sensors-23-07483-f011:**
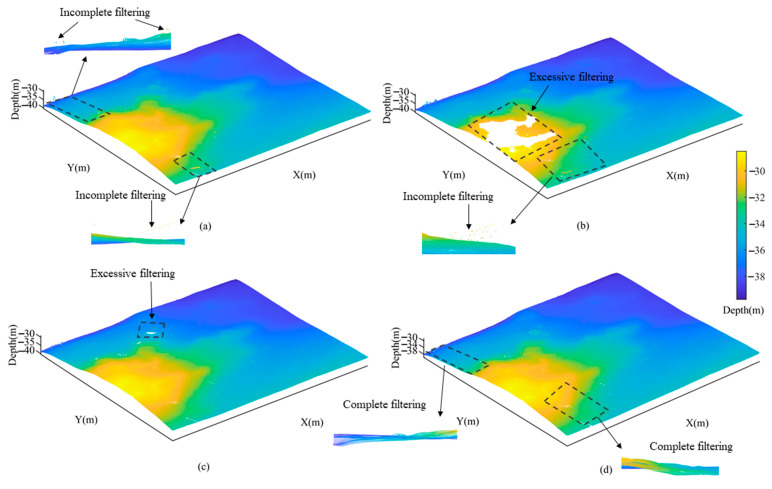
Comparison of filtering results in Area 2 using different methods. (**a**) CUBE; (**b**) TSF; (**c**) CSF; (**d**) CMF.

**Figure 12 sensors-23-07483-f012:**
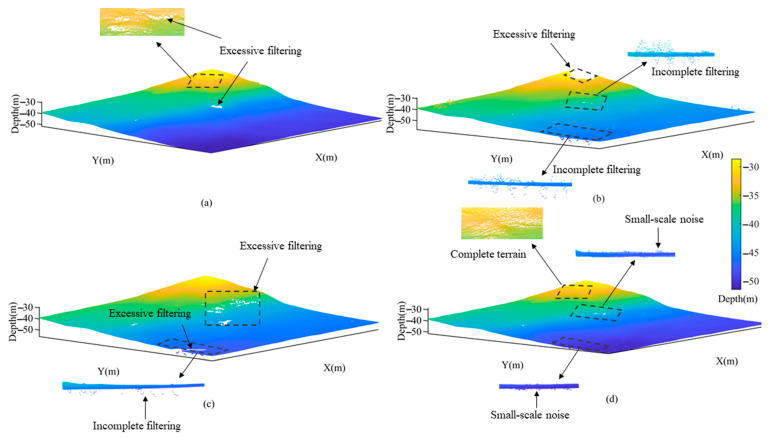
Comparison of filtering results in Area 3 using different methods. (**a**) CUBE; (**b**) TSF; (**c**) CSF; (**d**) CMF.

**Table 1 sensors-23-07483-t001:** Method accuracy and execution time at different voxel sizes.

*θ* Angle (°)	Voxel Size (m)	Type I Error (%)	Type II Error (%)	Time (s)
20	1 × 1 × 1	0.04	56.36	2.606
	2 × 2 × 2	1.21	57.54	2.563
	3 × 3 × 3	0.80	58.54	2.534
	4 × 4 × 4	0.66	59.62	2.467
	5 × 5 × 5	0.56	60.69	2.417
40	1 × 1 × 1	1.77	31.00	2.466
	2 × 2 × 2	1.73	42.23	2.416
	3 × 3 × 3	1.17	45.54	2.384
	4 × 4 × 4	1.03	49.46	2.378
	5 × 5 × 5	0.95	51.15	2.343
60	1 × 1 × 1	5.27	13.77	2.445
	2 × 2 × 2	4.54	21.85	2.399
	3 × 3 × 3	3.32	27.77	2.342
	4 × 4 × 4	2.95	35.69	2.326
	5 × 5 × 5	2.88	38.23	2.306

**Table 2 sensors-23-07483-t002:** Statistical analysis of mutual difference between HCI filtering and four other filtering methods.

Experimental Areas	Filtering Methods	Difference in Max Depth (m)	Difference in Min Depth (m)	Difference in Average Depth (m)	Difference in Variance	Difference in Standard Deviation
Area 1	CUBE	0.01	−2.07	−0.001	0.0106	0.0038
	TSF	2.11	−0.35	−0.001	0.0100	0.0036
	CSF	2.11	0.00	−0.010	−0.0147	−0.0053
	CMF	−0.01	−0.02	−0.003	0.0036	0.0012
Area 2	CUBE	0.00	0.06	0.000	0.0085	0.0017
	TSF	0.00	0.11	0.289	−1.2135	−0.2495
	CSF	0.00	0.00	0.003	0.0146	0.0029
	CMF	−0.04	0.00	0.000	0.0033	0.0007
Area 3	CUBE	0.01	0.08	−0.008	0.2042	0.0187
	TSF	6.56	0.11	0.219	−2.4851	−0.2330
	CSF	4.44	0.00	−0.099	0.1166	0.0107
	CMF	0.44	0.00	0.005	−0.0082	−0.0007

**Table 3 sensors-23-07483-t003:** A comparison of the accuracy between CMF and CUBE based on IHO criteria.

Experimental Areas	Filtering Methods	Total Points before Filtering	Total Points after Filtering	Total Points Meeting IHO Standards	Accuracy (%)
Area 1	CUBE	75,283	75,253	74,820	99.42
	CMF		75,017	74,858	99.78
Area 2	CUBE	59,799	59,781	59,649	99.78
	CMF		59,708	59,677	99.94
Area 3	CUBE	186,660	186,657	179,636	96.24
	CMF		185,464	185,123	99.81

## Data Availability

Not applicable.

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
