# Peer review of "A Novel Cone Model Filtering Method for Outlier Rejection of Multibeam Bathymetric Point Cloud: Principles and Applications"

_sensors, 2023, doi:10.3390/s23177483_

Round 1

Reviewer 1 Report

The authors proposed a cone model filtering (CMF) method based on an improved tornado algorithm. The occurrence of excessive filtering in areas with intricate underwater terrain is reduced by introducing virtual grids and voxel down-sampling. In this paper, the author collected sufficient bathymetric data, and employed three filtering approaches, namely CUBE, TSF and CSF to validate the efficacy of CMF from both qualitative and quantitative perspectives. In the quantitative analysis, the authors use HCI filter with high filtering accuracy as the basis for comparison, which has certain credibility. The latest standard published by the International Hydrographic Organization (IHO) is used to obtain the accuracy improvement of CMF method compared with CUBE method. The CMF method proposed by the authors shows good filtering effect under different terrain and has certain application value, but some issues are summarized in the following comments:

1. In Figure 1 (b), it is suggested to add subscripts of points p1 and p2 to the equation to make a clear distinction.

2. If the result of xmax-xmin in Equation (2) can divide GR exactly, does +1 still need to be followed? Will the extra mesh affect the accuracy or speed of the algorithm? What happens if the equation denotes rounding instead? Equation (3) has the same problems.

3. Please confirm whether Equation (6) is standard deviation or variance.

4. There is a big problem with the layout of this paper, many pages appear a lot of white space.

5. Line 293, why “the sum of θ angle and slope angle of steep terrain probably exceeded 90 degrees”. According to formula (7), their sum should be less than 90 degrees. The authors’ explanation fails to convince me.

6. Line 291, why “the impact of steep terrain on Type I and Type II errors increased significantly as the θ angle increased”. Please carefully analyze Figure 9 (c).

7. In Table 1, when the θ angle was 20 degrees and the voxel size was 1 × 1 × 1 m, Type I error was 0.04%. It is obviously different from the other data, despite the authors’ explanation, but did the authors run many tests on the situation to verify the correctness of the data?

8. In the analysis of Figure 9, the authors select the range of values of θ angle in the three types of terrain. However, in steep terrain, the θ angle can reach 20 degrees, and at this time, the Type I error and Total error are close to 10%, which is different from the value standard in the first two types of terrain. Is there a uniform and clear standard for selecting the range of θ angle?

none

Reviewer 2 Report

1.      There are multiple sentences in the article where there is excessive spacing between words, such as in line 36, 37, 48, and so on. The spacing between words should be consistent across paragraphs, and I think hyphens should be used.

2.      The content in the experimental result figures is blurry, and it is recommended to use vector graphics to display the content clearly.

3.    More recent works on array signal processing should be reviewed, e.g., 2D-DOA estimation for coherent signals via a polarized uniform rectangular array, in IEEE SPL.

4.      The formatting of the section titles from Section 1 to 5 should be consistent.

5.      To differentiate it from the original distance, I suggest that the left part of Equation '(z-zv)×tanθ' in Figure 1b should be displayed in red color.

6.      I think that in Eq. 4, "m" should be "n".

7.      The equation on line 228 should be aligned with the text.

The presensation can be improved further.

Reviewer 3 Report

The Abstract should be written in the present tense. So please correct:

Line 17: “proposes” instead of “proposed”

Line 18: “is” instead of “was”

Line 20: “are” instead of “were”

Line 23: “demonstrate” instead of “demonstrated”

Line 24: “removes” instead of “removed”

Line 26: “yields” instead of “yielded”

Line 34: “surveys” instead of “survey”

Line 35: please add “both” in front of “including to make it: “both including” since you refer in two kind of anomalies

Line 47: please delete the word “respectively”

Line 48 …up to… line 84: This is your previous work part. Every time that you mention in a direct way what a previous researcher has done then it is better to use Past Tense. For example in Line 50 instead of “Yang et al. propose” you should write “Yang et al. proposed”. Please correct appropriately all similar cases.

Line 93: Please make the title more descriptive (e.g., Description of the CMF methodology)

Line 109: “if the former is less than or equal…) This has to appear also as an equation to make it clear for the reader

Line 115: “to” instead of “of” and “denotes” instead of “denote”

Figure 6: in the depth color-bar please add “Depth (m)” like you have done in Figure 7

Line 235: “and again statistical filtering” Something is missing. Maybe you mean “and again statistical filtering was applied” please check it

Line 245: “by inequation”. This is wrong. Please make it “using equation”

Line 307: please delete “increases”

Line 367: please delete “the two”

Table 2: replace all “Difference of” with “Difference in”

Line 409: “some” instead of “Some”

Conclusions: Please use Present Tense to summarize your work. For example: in Line 422 use “This paper proposes” instead of “This paper proposed”. Please correct appropriately all similar cases.

Moderate editing of English language required. 

Round 2

Reviewer 1 Report

none

none

Reviewer 2 Report

There is a lack of comparison with the state-of-the-art approaches. Simulations should be designed in the revised manuscript.

It's OK.
